# Magnetically propagating Hund's exciton in van der Waals antiferromagnet NiPS₃

W. He [1] ✉, Y. Shen [1], K. Wohlfeld [2], J. Sears[1], J. Li[3], J. Pelliciari [3], M. Walicki [2], S. Johnston[4,5], E. Baldini [6], V. Bisogni [3], M. Mitrano [7] & M. P. M. Dean [1] ✉

Magnetic van der Waals (vdW) materials have opened new frontiers for realizing novel many-body phenomena. Recently NiPS₃ has received intense interest since it hosts an excitonic quasiparticle whose properties appear to be intimately linked to the magnetic state of the lattice. Despite extensive studies, the electronic character, mobility, and magnetic interactions of the exciton remain unresolved. Here we address these issues by measuring NiPS₃ with ultra-high energy resolution resonant inelastic x-ray scattering (RIXS). We find that Hund's exchange interactions are primarily responsible for the energy of formation of the exciton. Measuring the dispersion of the Hund's exciton reveals that it propagates in a way that is analogous to a double-magnon. We trace this unique behavior to fundamental similarities between the NiPS₃ exciton hopping and spin exchange processes, underlining the unique magnetic characteristics of this novel quasiparticle.

Two-dimensional (2D) vdW materials provide an ideal platform for combining strong electronic correlations, low-dimensional magnetism, and weak dielectric screening to realize novel electronic quasiparticles and functionality[1–3]. Recent years have seen the identification of excitons in a host of closely related vdW compounds such as NiPS₃, CrSBr, NiI₂, and MnPS₃[4–7]. Within this family, the NiPS₃ exciton exhibits several fascinating properties, including strong interactions between the exciton lifetime and magnetic order[4], thickness-dependent properties in the few-layer-limit[8–10], coupling between magnetism and exciton polarization[8,11,12], and unconventional exciton-driven metallic behavior[13]. These observations suggest that excitons in magnetic vdW materials such as NiPS₃ might have fundamentally different character from other types of excitons, such as the Frenkel, Wannier, and Hubbard varieties. Frenkel and Wannier excitons form via Coulomb interactions between electrons and holes in different Bloch states and propagate according to the detailed form of the band-structure and electron-hole attraction[14]. Hubbard excitons, on the other hand, form from strongly correlated many-

body states, and their propagation is expected to involve the scattering of spin waves[15,16].

In previous works, the NiPS₃ exciton has been described as a "Zhang-Rice" mode[4]. This terminology derives from studies of cuprate superconductors, and refers to a specific form of hybridized wavefunctions that have one hole on the transition metal (Ni) site and one hole on the ligand (S) site and describes the "Zhang-Rice exciton" as a transition from a high-spin triplet to a low-spin singlet[17]. However, the way the exciton changes with applied magnetic field has been argued to be incompatible with this picture[18]. A Zhang-Rice scenario also does not address why the exciton has such a narrow linewidth[8,11,19]. The unsettled and possibly unconventional exciton electronic character suggests that the exciton may also propagate in an exotic manner different to regular excitons, but, to date, this has never been measured. Here, we use ultra-high energy resolution RIXS to directly detect the NiPS₃ exciton momentum dispersion and discover it propagates magnetically in a similar way to the double-magnon excitation. Through detailed analysis of the exciton wavefunction, we further

¹Department of Condensed Matter Physics and Materials Science, Brookhaven National Laboratory, Upton, NY 11973, USA. ²Institute of Theoretical Physics, Faculty of Physics, University of Warsaw, Warsaw PL-02093, Poland. ³National Synchrotron Light Source II, Brookhaven National Laboratory, Upton, NY 11973, USA. ⁴Department of Physics and Astronomy, The University of Tennessee, Knoxville, TN 37996, USA. ⁵Institute of Advanced Materials and Manufacturing, The University of Tennessee, Knoxville, TN 37996, USA. ⁶Department of Physics, The University of Texas at Austin, Austin, TX 78712, USA. ⁷Department of Physics, Harvard University, Cambridge, MA 02138, USA. ✉e-mail: whe1@bnl.gov; mdean@bnl.gov

reveal the different interactions involved in its formation and establish that its primary character is that of a Hund's exciton, distinct from the Zhang-Rice and other scenarios.

## Results

### Electronic character of the exciton

We start by measuring the incident energy dependence of the Ni $L_3$-edge RIXS spectrum of NiPS$_3$ to identify the different spectral features present (see Fig. 1a and Methods). The most intense peaks centered around 1.0, 1.1, and 1.7 eV energy loss are $dd$ excitations in which electrons transition between different Ni $3d$ orbitals. A remarkably sharp (almost resolution limited) peak is apparent at an energy loss matching the known energy of the exciton at 1.47 eV. This excitation resonates strongly at 853.4 eV and is well separated from other $dd$ and the higher energy charge-transfer excitations. We identify this feature as the NiPS$_3$ exciton, consistent with previous reports[4].

To facilitate our understanding of the exciton and its interplay with magnetism, we constructed an effective NiS$_6$ cluster model representing NiPS$_3$ (See Supplementary Note 1a). Our model includes Coulomb repulsion, Hund's coupling, crystal field, and Ni-S and S-S hopping. As explained in the Methods section, the rich spectrum, including the detailed splitting of the two $dd$-excitations at 1.0 and

1.1 eV allows us to obtain a well-constrained model Hamiltonian for NiPS$_3$ (see Fig. 1b, c, Supplementary Note 1b, and Supplementary Fig. 2).

To better understand the nature of the exciton, we plot several expectation values describing the NiPS$_3$ wavefunction in Fig. 1d–f, which reveal that NiPS$_3$ has dominant Hund's rather than Zhang-Rice character. The plotted expectation values include the hole occupations of the Ni $3d$ and ligand orbitals and the weights of the $d^8$, $d^9\underline{L}$, and $d^{10}\underline{L}^2$ configurations ($\underline{L}$ stands for a ligand hole) that make up each state. We also calculate the expectation values for the total spin operator squared $\hat{S}^2$, which, for two holes, has a maximum value of $S(S+1) = 2$. The ground state is close to this pure high spin state; while the spin is reduced in the exciton. We therefore confirm that the exciton is dominantly a triplet-singlet excitation. In the Zhang-Rice scenario, the leading character of the ground state would be $d^9\underline{L}$. The dominant component is, in fact, $d^8$ revealing that the state has dominant Hund's character.

The ground state and the exciton wavefunctions obtained from our model are illustrated in Fig. 1g, h and are described in detail in Supplementary Note 2. We see that substantial charge redistribution occurs during exciton formation, which is partly crystal field and partly a Ni-S charge transfer in nature (see Fig. 1d, e). As explained later in the discussion section, the fact that the ground states have dominant Ni

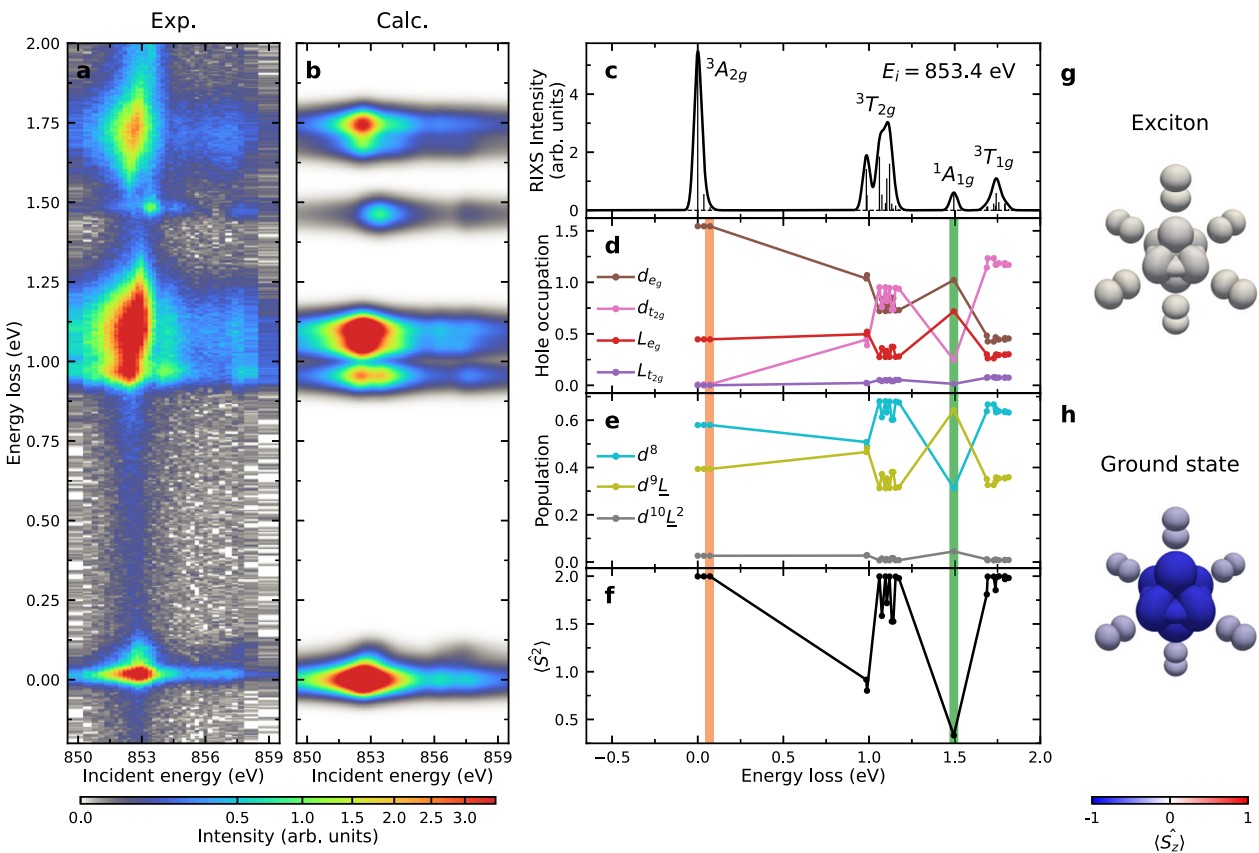

**Fig. 1 | Electronic character of the NiPS$_3$ exciton. a** RIXS intensity map as a function of incident photon energy through the Ni $L_3$ resonance. The exciton is visible at an energy loss of 1.47 eV and reaches a maximum intensity at an incident energy of 853.4 eV. These data were taken at 40 K with $\pi$-polarized x-rays incident on the sample at $\theta = 22.6°$ and scattered to $2\Theta = 150°$. **b** RIXS calculations for NiPS$_3$ that capture the energy and resonant profile of the $dd$-transitions and exciton in the material. **c** Calculated unbroadened RIXS intensity (vertical lines) and broadened RIXS spectra (solid curve) at the main resonant incident energy of the exciton peak (i.e., $E_i = 853.4$ eV). **d–f** Description of the ground and excited states in NiPS$_3$. **d** shows the hole occupations of Ni $3d$ (denoted by $d$) and ligand (denoted by $L$) orbitals. **e** displays probabilities of having $d^8$, $d^9\underline{L}$, and $d^{10}\underline{L}^2$ configurations.

**f** gives the expectation value of the total spin operator squared $\langle \hat{S}^2 \rangle$. The orange (green) vertical lines in **d–f** indicate the energy for the double-magnons (excitons). **g, h** Wavefunction illustrations extracted from **b** for **g** the exciton and **h** the ground state. The size of each orbital ($3d$ for the central Ni site and $3p$ for the six neighboring S sites) is proportional to its hole occupation. The color represents the expectation value of the spin operator along the z axis $\langle \hat{S}_z \rangle$, again calculated separately for the Ni and S states. Therefore, the change in spin state and the partial transfer of holes involved in the exciton transition is encoded in the change in color and size of orbitals, respectively. We represent the ground state by only the down-spin configuration, omitting the up-spin and spin-zero elements of the triplet.

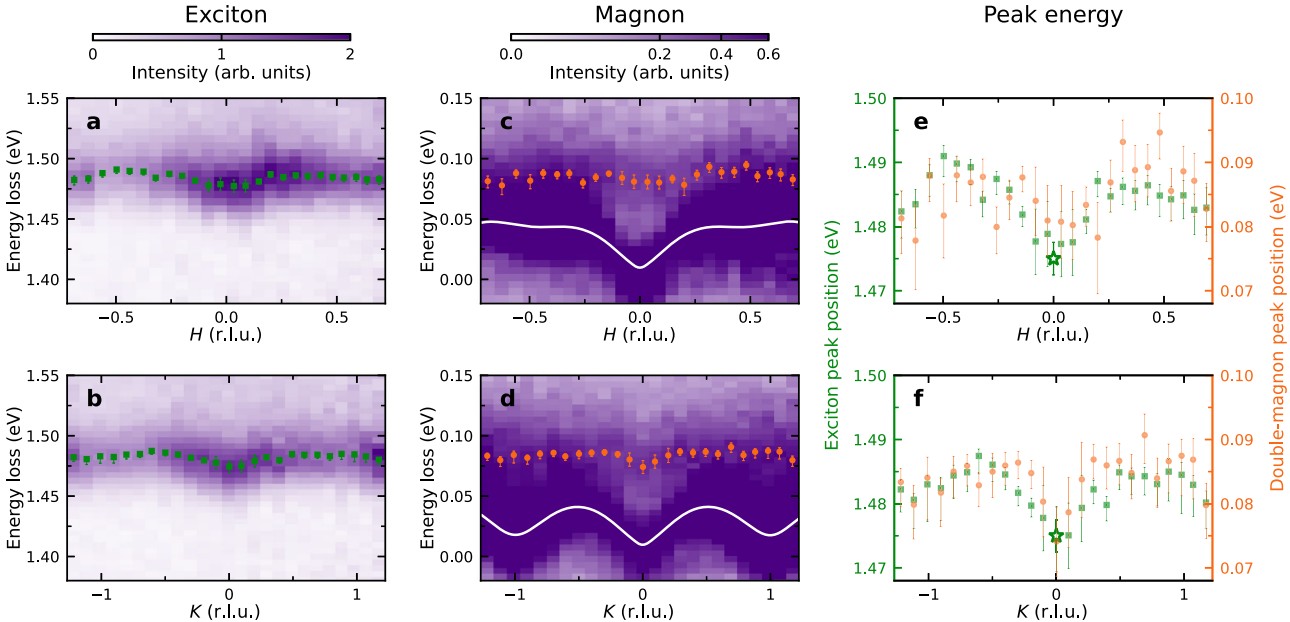

**Fig. 2 | Low temperature exciton dispersion and comparison with double-magnons. a, b** RIXS intensity maps as a function of the *H* and *K* in-plane momentum transfer, respectively, with an energy window chosen to isolate the exciton dispersion. The overlaid green squares mark the peak positions of the exciton. **c** and **d** show the low energy dispersion at equivalent momenta with the observed inelastic feature, including magnons (white lines) and double-magnons (orange circles). **e** and **f** show that the exciton and double-magnon have similar dispersion character (rather than dominant Zhang-Rice character) plays a leading role in the energy for exciton formation.

with an energy offset of ~1.4 eV. All the measurements were taken at $T = 40$ K using $\pi$-polarized incident x-rays at an incident energy of 853.4 eV corresponding to the exciton resonance. The asterisks in **e** and **f** denote the reported exciton energy from optical measurements[4,8,11] with error bars from our instrument energy calibration (one standard deviation). All other error bars are 1-$\sigma$ confidence intervals evaluated from the fitting as explained in the Methods. Detailed linecuts showing the fitting are provided in Supplementary Figs. 3–5.

## Exciton dispersion

Having clarified the character of the exciton, we study its propagation by tuning the incident x-ray energy to the exciton resonance at 853.4 eV and mapping out the in-plane dispersion with high energy resolution (Fig. 2a, b). The two high-symmetry in-plane reciprocal space directions both exhibit a small upward dispersion away from the Brillouin zone center with similar bandwidths of ~15 meV. This non-zero dispersion suggests that the exciton excites low-energy quasiparticles as it propagates through the lattice. We consequently mapped out the low energy excitations in Fig. 2c, d. The strongest feature is the magnon, which was found to be consistent with the prediction based on prior inelastic neutron scattering measurements[20,21] (the white line, see Methods). Intriguingly, we also found another broad low-energy dispersive feature at an energy scale roughly twice as large as the magnon peak (the orange dots in Fig. 2c, d). As we justify in detail later, the observed low-energy feature, in fact, corresponds to "double-magnon" excitations, which is a process in which two spins are flipped on each site making up the excitation to create a pair of magnons with the same spin. By fitting the exciton and double-magnon energy, we see that these two excitations show similar dispersion despite their drastically different energy scales (Fig. 2e, f), indicating that the exciton propagates in a way that is similar to the double-magnon. Since the dispersive effects are subtle, we confirmed the calibration of the spectra by verifying that the magnon energy exactly reproduces the dispersion obtained in prior inelastic neutron scattering experiments[20,21]. We similarly confirmed that the Brillouin zone center exciton energy is consistent with values from optical measurements[4,8,11].

## Temperature dependence

To substantiate the identity of the low energy excitations we observe in $NiPS_3$, we need to consider the RIXS cross-section. Since RIXS is a photon-in photon-out scattering process with each photon carrying one unit of angular momentum, it can couple to processes involving either zero, one, or two spin flips[22–26]. While the spin-flip processes are necessarily magnetic, the zero spin flip process can either correspond to a phonon or to a so called "bi-magnon" in which two magnons with opposite spins are created on neighboring sites to make an excitation with a net spin of zero. The most straightforward way to distinguish magnetic and non-magnetic excitations is to measure the dispersion above the Néel order temperature of $T_N = 159$ K, as presented in Fig. 3. Above $T_N$ the exciton remains visible, but it becomes weaker and more diffuse compared to the data at 40 K (see Fig. 3a and the linecuts in Supplementary Fig. 10). Consequently, no dispersion is detectable. The double-magnon peak that was observed at 40 K is replaced by a diffuse, over-damped tail of intensity, contrary to what would be expected for a phonon and corroborating its magnetic origin (see Fig. 3b and the linecuts in Supplementary Fig. 9). The residual intensity arises from short-range spin fluctuations, which are expected to persist well above $T_N$ in quasi-2D magnets as long as the thermal energy scale is well below the energy scale of the magnetic interactions[27]. We also note that optical phonons in the 70–100 meV energy range in $NiPS_3$ are known to be minimally-dispersive[28], which again suggests a magnetic origin.

## Identifying double-magnons through their resonant profile

Having established a magnetic origin for the dispersion, we can use the energy-dependent resonant profile of RIXS to distinguish different magnetic processes and substantiate our assignment of the low-energy feature as the double-magnon. Figure 4a plots the energy dependence of the measured double-magnon feature alongside the magnon and the exciton. We compare this with calculations of the RIXS cross-section based on our model in Fig. 4b, which includes processes involving zero, one, and two spin flips denoted by $\Delta m_S = 0$, 1, and 2, respectively, as well as coupling to the exciton. $\Delta m_S = 0$ reflects the cross section for either elastic scattering or a bi-magnon, $\Delta m_S = 1$

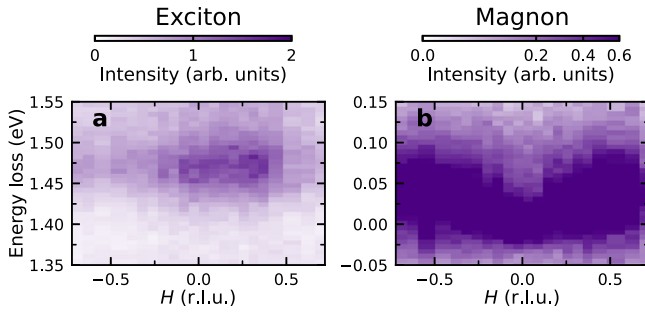

**Fig. 3 | High temperature exciton dispersion and comparison with double-magnons above the Néel transition.** The RIXS intensity maps with energy window chosen to isolate the excitons **a** and magnons/double-magnons **b** as a function of in-plane momentum transfer $H$ measured in the ($H0L$) scattering plane. All the measurements were taken at $T = 190$ K with linear horizontal $\pi$ polarization of the incident x-rays at the resonant energy of 853.4 eV for excitons. The same data are provided as linecuts in Supplementary Figs. 9 and 10.

corresponds to a magnon, and $\Delta m_S = 2$ corresponds to the double-magnon. Notably, since the double-magnon involves an exchange of two units of spin angular momentum, RIXS is especially suitable for detecting this process. The main resonance around 853 eV shows partial overlap between the double-magnon and bi-magnon resonance, so it does not clearly distinguish between them. However, the presence of the excitation at the satellite resonance at 857.7 eV is only compatible with a $\Delta m_S = 2$ double-magnon process. We therefore suggest that the dominant character of the excitation around 80 meV is that of double-magnons substantiating our spectral assignment, although we cannot exclude a small sub-leading contribution. Our assignment is also supported by RIXS studies of NiO where the double-magnon was also found to be much more intense than the bi-magnon[22,24,26,29]. The satellite resonance is generated by the exchange part of the core-valence Coulomb interaction on the Ni site. This same interaction creates the double-magnon because the angular momentum state of the core-hole needs to vary in the intermediate state in order to allow two subsequent spin-flip processes to occur in the photon absorption and emission processes. The core-valence exchange interaction facilitates this change in the core-hole state by mixing of the core-hole and valence eigenstates. This conclusion is also borne out in studies of NiO where the same process occurs[22,24,26,29].

## Discussion

In this work we use high resolution RIXS to assess the formation and propagation of the excitonic state of NiPS$_3$. By combining RIXS and ED calculations, we reveal that the primary mechanism behind the exciton formation is the Hund's interaction. As illustrated in Fig. 1d–h the exciton forms from a ground state with dominant $d^8$ character and involves significant charge transfer and crystal field changes. As such, the state we identify is quite different from prior descriptions of a Zhang-Rice exciton[4]. As we discuss later, the distinction between these models is crucial as it corresponds to a different majority component of the wavefunction, different interactions playing the leading role in the exciton energy, and the possibility of realizing a model with physically reasonable parameters. These issues will be central to efforts to manipulate the exciton energy and cross section. We found that the difference in the prior identification of the exciton arises from using an under-constrained model. If one considers just the exciton energy and assumes that Hund's coupling can take any value, there are a range of different Hund's interactions and charge transfer energy parameters that predict a 1.47 eV exciton. If one adds a further constraint that the 1.0, 1.1, and 1.7 eV $dd$-excitations must be reproduced within an accuracy similar to their width, properly constrained solutions can be identified (see Supplementary Note 1f). Importantly, the solution

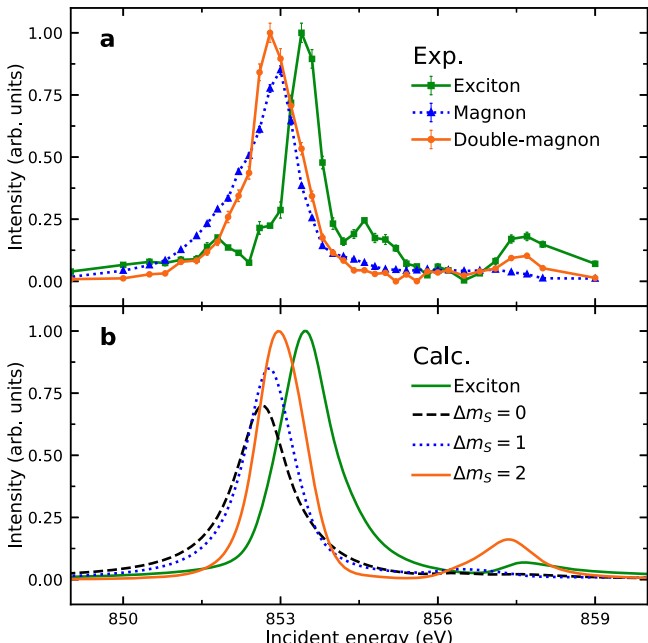

**Fig. 4 | Resonance behavior of magnons, double-magnons, and excitons. a** The measured RIXS spectral weights of the magnon, double-magnon and exciton extracted by fitting experimental RIXS spectrum for each incident energy. Error bars represent one standard deviation. Data were taken at 40 K at a scattering angle of $2\Theta = 150°$ and an incident angle of $\theta = 22.6°$. **b** The calculated RIXS spectral weights of the exciton and low-energy zero-, one-, and two-spin-flip transitions ($\Delta m_s = 0, 1, 2$) as a function of incident energy. The curves in both panels are scaled for clarity.

found here also yields a physically reasonable value for Hund's coupling $J_H = 1.24$ eV corresponding to 87% of the atomic value. This is relevant because the pure triplet and singlet Zhang-Rice components of the wavefunctions are energetically split by rather weak Ni-S exchange processes, so it is difficult to justify the 1.47 eV energy scale of the exciton within a model with dominant Zhang-Rice character. In Ref. 4 an unphysically large Hund's coupling corresponding to 120% of the atomic value was required.

Based on the wavefunction extraction performed in this study, we can determine which electronic interactions play the leading role in the exciton energy. To do this, we factorized the wavefunctions into the singlet and triplet components of the $d^8$, $d^9\underline{L}$, and $d^{10}\underline{L}^2$, configurations as described in Supplementary Note 2. We can then compute the contributions of Hund's, charge transfer, and crystal field to each of these components. We find that the primary contribution to the exciton energy comes from singlet-triplet splitting of the $d^8$ component of the wavefunction, which means that it is best thought of as a Hund's exciton since its energy of formation is mostly driven by Hund's exchange. Our exciton character derived here, also retains partial magnetic character coming from a sizable contribution of states beyond the three $d$-electron, Zhang-Rice, and ligand, singlet states. This means that the exciton is expected to vary with applied magnetic field compatible with recent observations[18]. It also implies that future efforts to realize similar symmetry excitons with different energies should target means to modify the on-site Ni Hund's exchange coupling and not the Ni-S exchange processes that would be the leading contribution to the energy of a Zhang-Rice exciton.

Our detection of exciton dispersion in NiPS$_3$ proves that the exciton is an intrinsic propagating quasiparticle and excludes prior suggestions that the exciton might be a localized phenomenon associated with defects[30]. The most common form of exciton propagation in weakly correlated transition metal chalcogenides involves

excitations that are composed of bound pairs of specific Bloch states[31,32]. The NiPS$_3$ exciton is quite different from the more conventional excitons since it is bound by the local Hund's interactions described previously, rather than long-range Coulomb attraction. Recent calculations that work for many more conventional excitons, indeed fail to capture our measured exciton dispersion[33]. We also note that the two holes in the Ni $e_g$ manifold represent what is in some sense the simplest way to realize a triplet-singlet excitation. Consequently, NiPS$_3$ has relatively few excitations compared to other materials and the exciton is energetically well separated from other transitions, such that it interacts exclusively by magnons and not other types of excitations. The exciton character is also dominated by processes that rearrange spins on the Ni site, rather than moving charge between more extended states, which would tend to reduce the coupling of the exciton to phonons. These factors may contribute to the long lifetime and narrow linewidth of the exciton.

A key result of this study is that the NiPS$_3$ exciton propagates like the double-magnon, even though the average energies of the exciton and double-magnon differ by more than one order of magnitude. This remarkable similarity can be understood by analyzing the exchange processes involved in the motion of the quasiparticles. We start by considering the spin-superexchange processes involved in exciton motion, finding that the exciton can swap its position with a spin (see Supplementary Note 4b for details). Consequently, when the exciton moves through the antiferromagnetic lattice, it generates a string of misaligned spins. Given that the exciton appears to propagate freely, we should consider processes that heal the misaligned spins in the wake of the exciton, which leads to the image in Fig. 5a that illustrates the spin flips involved in exciton motion. Similar considerations can be applied to the motion of a double-magnon as shown in Fig. 5b. Importantly, both exciton and double-magnon motion involve four spin exchanges. If we consider the sequence of different overlap integrals involved on the Ni and ligand states, the amplitude of the exciton hopping and double-magnon exchange processes are expected to be quite similar (see Supplementary Notes 4a and 4c). These considerations help us rationalize the similarities in the propagation of

the two quasiparticles. A simple empirical tight-binding model fit to the exciton dispersion (see Supplementary Note 5) reveals that the third nearest neighbor interaction is the leading term in determining the exciton dispersion, consistent with the third nearest neighbor spin exchange being the dominant term in the spin Hamiltonian[20,21]. We note in passing that a similar picture for a well-known free *single* magnon propagation in an antiferromagnet would require generating two spin exchanges[34].

The coupling between the exciton and magnetism might be relevant to why the 1.47 eV exciton feature is visible in optics experiments. Since the exciton involves transitions between $d$-orbitals, it is expected to be nominally optically forbidden in a centrosymmetric crystal due to dipole selection rules. However, these rules can be lifted by perturbations that break the Ni-site symmetry, which include exciton-spin or exciton-lattice interactions. The fact that there is a strong change in the optical cross-section through $T_N$[35], whereas the RIXS is only modestly broadened, also supports the interpretation that exciton-spin interactions play a key role in the optical cross-section.

Overall, our measurements reveal a Hund's excitonic quasiparticle in NiPS$_3$ that propagates in a similar manner to a two-magnon excitation. The coming years will likely see further instrumental developments that allow RIXS, and exciton microscopy, measurement of NiPS$_3$ to be extended to the ultrafast pump-probe regimes[36,37]. We believe that this has outstanding potential for understanding new means of using magnetic Hund's excitons to realize new forms of controllable transport of magnetic information.

Note added: We note that a similar study has been performed on nickel dihalide materials[38]. These results also identify a Hund's exciton with a magnetic propagation mechanism, highlighting the widespread applicability of our conclusions.

## Methods
### Sample information
NiPS$_3$ bulk single crystal samples were procured from 2D Semiconductors, which synthesized the crystals by the chemical vapor transport method. The full unit cell of NiPS$_3$ has a monoclinic symmetry (Space Group $C2/m$, #12) with lattice parameters $a = 5.8$ Å, $b = 10.1$ Å, $c = 6.6$ Å, and $\beta = 107.0°$. We adopted this monoclinic-unit-cell convention, and index reciprocal space using scattering vector $\boldsymbol{Q} = (H, K, L)$ in reciprocal lattice units (r.l.u.). Therefore, the reciprocal lattice vector $\boldsymbol{c}^*$ is perpendicular to the $ab$ plane.

Ni$^{2+}$ ions in NiPS$_3$ lie on a honeycomb lattice in the $ab$ plane and form with ABC-type stacking of the layers. Such stacking breaks the three-fold rotational symmetry of the monolayer structure which can be detected by measuring structural Bragg peaks such as $(0, 2, 4)$. NiPS$_3$ is prone to characteristic twinning involving three equivalent domains rotated by 120° in the $ab$ plane[20,39]. Laboratory single-crystal x-ray diffraction measurements confirmed the presence of three twin domains in our samples. Therefore, measured quantities should be a weighted average over the three twin domains. We included these domain-averaging effects in the ED and magnon energy calculations. The apparent similarity in the measured dispersion between the two distinct directions (i.e., antiferromagnetic across the zig-zag chain and ferromagnetic along the chain) can be ascribed to the domain averaging effect due to the presence of structural twinning in the measured sample.

NiPS$_3$ orders magnetically below a Néel temperature of $T_N = 159$ K[39]. The magnetic unit cell is the same as the structural unit cell. It has a collinear magnetic structure consisting of antiferromagnetically coupled zigzag chains.

### RIXS measurements
Ultra-high-energy-resolution RIXS measurements were performed at the SIX 2-ID beamline of the National Synchrotron Light Source II[40]. The surface normal of the sample was $\boldsymbol{c}^*$ axis (i.e., $L$ direction). The in-

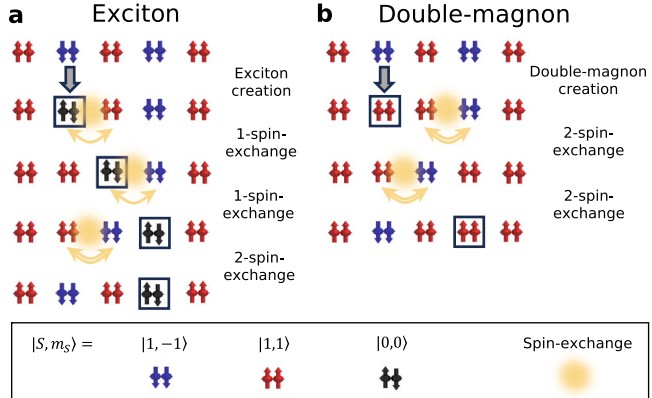

**Fig. 5 | Illustration of exciton and double-magnon propagation based on perturbation theory.** The top row represents the antiferromagnetic background and subsequent rows show the time evolution of the state. **a** After the singlet $|0,0\rangle$ exciton forms (second row from the top) it exchanges spin with neighboring sites such that it moves while flipping spins and breaking magnetic bonds; free propagation to the next nearest neighbor site (bottom row) is possible after four spin exchanges, involving up to four magnons created in the intermediate state (middle rows). **b** After the double-magnon excitation is created on the same site (second row from the top), it can freely move to the next nearest neighbor (bottom row) by four spin exchanges and exciting four magnons in the intermediate state (middle rows). The similarities between the propagation in **a**, **b** rationalize the experimentally observed similar dispersion relation of the exciton and double-magnon. These processes are mediated by the different spin-exchange interactions, with the third nearest neighbor exchange process playing the leading role.

**Table 1 | Full list of parameters used in the AIM calculations**

| 10Dq | $\epsilon_p$ | $V_{pd\sigma}$ | $V_{pp\sigma}$ | $F^0_{dd}$ | $F^2_{dd}$ | $F^4_{dd}$ | $F^0_{LL}$ | $F^2_{LL}$ | $F^4_{LL}$ | $U_{dL}$ | $F^0_{dp}$ | $F^2_{dp}$ | $G^1_{dp}$ | $G^3_{dp}$ | $\zeta_i$ | $\zeta_n$ | $\zeta_c$ |
|---|---|---|---|---|---|---|---|---|---|---|---|---|---|---|---|---|---|
| 0.42 | 7.5 | 0.95 | 0.8 | 7.88 | 10.68 | 6.68 | 0.46 | 2.59 | 1.62 | 1 | 7.45 | 6.56 | 4.92 | 2.80 | 0.083 | 0.102 | 11.4 |

Units are eV.

plane orientation was determined by Laue diffraction. Then the pre-aligned sample was cleaved with Scotch tape in air to expose a fresh surface and immediately transferred into the RIXS sample chamber. Ni $L_3$-edge RIXS measurements were taken with linear horizontal ($\pi$) polarization with a scattering plane of either ($H0L$) or ($0KL$). The main resonance energy (around 853 eV) is common for Ni-containing compounds[41–43] but different from the previous report[4]. This difference comes from the absolute energy calibration of the beamline and does not affect the RIXS measurements and interpretations, which depend only on the relative changes. The spectrometer was operated with an ultrahigh energy resolution of 31 meV full-width at half-maximum (FWHM). The temperature of the sample was kept at $T = 40$ K except for the temperature-dependent measurements. Since the interlayer coupling in NiPS$_3$ is relatively weak, the dispersion measurement was taken at a scattering angle of $2\Theta = 150°$ while varying the incident angle of the x-rays. A self-absorption correction[44] was applied to the RIXS spectra, which, however, does not affect peak positions.

### Fitting of the RIXS spectra
In order to quantify the dispersion of the excitons and double-magnons, we fitted these peaks in the measured RIXS spectra to extract their peak positions. Although, in principle both the exciton and double-magnon can contain detailed substructure, we found that both features in our spectra can be accurately fit with simple peak shapes.

In the double-magnon region, we used a Gaussian function for the elastic peak, a damped harmonic oscillator (DHO) model for both the magnon and double-magnon peaks, and a constant background. The width of the elastic peak was fixed to the energy resolution, which was determined by a reference measurement on a multilayer heterostructure sample designed to produce strong elastic scattering. The DHO equation for RIXS intensity $S(\boldsymbol{Q}, \omega)$ as a function of $\boldsymbol{Q}$ and energy $\omega$ is

$$S(\boldsymbol{Q},\omega) = \frac{\omega\chi_Q}{1 - \exp(-\omega/k_B T)} \cdot \frac{2z_Q f_Q}{\left(\omega^2 - f_Q^2\right)^2 + (\omega z_Q)^2} \quad (1)$$

where $f_Q$ is the undamped energy, $\chi_Q$ is the oscillator strength, $z_Q$ is the damping factor, $k_B$ is the Boltzmann constant, and $T$ is temperature. This DHO was then convoluted with a resolution function (a Gaussian function with peak width fixed to the energy resolution) to describe the magnons and double-magnons. We used the fitted value of $f_Q$ to represent the magnon or double-magnon peak energy and used $z_Q$ to characterize the peak width.

In the exciton region, we resolved both the main exciton and an additional exciton sideband separated by ~40 meV as shown in Supplementary Fig. 3 (e.g., the spectrum at $K = -1.22$ r.l.u.), consistent with a previous report[4]. Therefore, we fitted the data with two Voigt peaks with a third order polynomial background. The width of the Gaussian component was fixed to the energy resolution, and we constrained the widths of these two peaks to be the same. For the scans where the minor peak is not obvious, we further fixed the spacing between these peaks to the average value of 40 meV obtained from other scans.

### Exact diagonalization RIXS calculations
Our NiPS$_3$ data were interpreted using standard ED methods for computing the RIXS intensity[23]. The Kramers-Heisenberg formula for the cross-section was used. This is derived by treating the

interaction between the photon and the material within second-order perturbation theory (as is required for scattering via an intermediate state resonance). We use the polarization-dependent dipole approximation for the photon absorption and emission interactions and simulate the presence of a core hole in the intermediate state with a core hole potential. In strongly correlated insulators like NiPS$_3$, accurate treatment of the electron–electron interactions is particularly important and the brief presence of a core hole means that local processes dominate the scattering. These factors mean that cluster approximations are particularly appropriate and are widely used for this reason[23]. We therefore perform calculations for a NiS$_6$ cluster, which can be projected onto Anderson impurity model (AIM) with essentially no loss of accuracy. As explained in detail in Supplementary Note 1, we were able to extract a well-constrained effective model for NiPS$_3$ from the data. This was used to generate Fig. 1b based on the parameters specified in Table 1. The detailed definitions of these parameters can be found in Supplementary Note 1.

Since the ED method employed involves directly computing wavefunctions, the model can be used to extract the wavefunctions plotted in Fig. 1g, h as outlined in Supplementary Note 2. All the calculations were done using the open-source software EDRIXS[45].

### Magnetic cross-section calculations
With the validated model in hand, it can be used to compute the resonance profile of the magnetic cross-section. A small Zeeman interaction was applied to the total spin angular momentum of the system, serving as the effective molecular magnetic field in the magnetically ordered state. The initially degenerate ground triplet consequently splits into three levels separated based on the spin state. After diagonalizing the Hamiltonian matrix, we used the Kramers-Heisenberg formula in the dipole approximation to calculate the incident-energy dependent RIXS cross section for transitions between the different elements of the triplet. The experimental geometry was explicitly included in the calculations. Three-fold twinning was also accounted for, which in fact has no effect on the final results due to the preserved cubic symmetry. We also note that these results are independent of the magnitude of the applied effective molecular magnetic field since this interaction is much smaller than the splitting between the ground state triplet and the lowest energy $dd$-excitation.

## Data availability
The RIXS data generated in this study have been deposited in the Zenodo database under access code 1079107[46].

## Code availability
The calculations in this study were performed using the open source code EDRIXS[45].

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

## Acknowledgements

We thank Sasha Chernyshev, Andy Millis, and Tatsuya Kaneko for fruitful discussions. Work performed at Brookhaven National Laboratory was supported by the U.S. Department of Energy (DOE), Division of Materials Science, under Contract No. DE-SC0012704. This work was partly supported by the U.S. DOE Office of Science, Early Career Research Program. Work performed at Harvard University (data interpretation and paper writing) was supported by the U.S. DOE, Division of Materials Science, under Contract No. DE-SC0012704. Theoretical calculations by K.W. was supported by the Excellence Initiative of the University of Warsaw (the 'New Ideas' programme, decision No. 501-D111-20-2004310). E.B. was supported by the United States Army Research Office (W911NF-23-1-0394). S. J. acknowledges support from the National Science Foundation under Grant No. DMR-1842056. This research used beamline 2-ID of the National Synchrotron Light Source II, a U.S. DOE Office of Science User Facility operated for the DOE Office of Science by

Brookhaven National Laboratory under Contract No. DE-SC0012704. We also acknowledge resources made available through BNL/LDRD19-013.

## Author contributions

The project was conceived by M.P.M.D. W.H., Y.S., J.S., J.L., J.P., E.B., V.B. and M.P.M.D. performed the measurements. W.H., Y.S., K.W., M.W., S.J. E.B., M.M. and M.P.M.D. interpreted the data and performed the calculations. The paper was written by W.H., Y.S. and M.P.M.D. with input from all co-authors.

## Competing interests

The authors declare no competing interests.
