## [Peer Review File · Nature Communications]

REVIEWER COMMENTS

Reviewer #1 (Remarks to the Author):

The authors present a detailed RIXS investigation of NiPS₃, a material interesting due to the existence of ingap excitons. They find that, contrary to previous investigations the exciton can be related to a Hund's rule singlet-triplet transition instead of a Zang-Rice Singlet excited state.

The paper is well written and the experimental investigations combined with theory are convincing.

I have very little comments to the paper and can recommend it for publication. A few minor comments would relate to Fig 1.

1) It would be nice to add the symmetry labels for the local states to the states as one generally sees for these local atomic multiplets

2) The picture of Fig 1.h shows the local "wave-function" or hole charge density. The colour indicates the spin of the state. It is not clear how this picture is made. At the same time I do not understand the change in spin between the Ni site and the O site. If these are the results of a Ni-Ligand cluster calculation I would expect the spin to be all parallel. (for the ground-state).

Reviewer #2 (Remarks to the Author):

The authors uses RIXS to study the nature of the exciton in the two-dimensional vdW magnetic material NiPS₃, claimed in an earlier study to arise from the transition between Zhang-Rice triplet and singlet states. The interpretation of the RIXS spectra is aided by a cluster model calculation, based on which it is claimed that the exciton is formed primarily from Hund's exchange, distinct from ZR and other scenarios. Additionally, the authors claim that the Hund's exciton propagates in a way that is analogous to a "double magnon". After reading the manuscript, I am left with many questions as listed below, which I hope the authors can provide in the revised version of their manuscript.

A. Since the charge transfer energy is positive, it is clear that the ground state has a larger weight for d^8 relative to d^9L-1 , in both this work and Ref. [4]. The difference between these two works in terms of the hole occupation on the Ni Site is 1.26 versus 1.08. To me, this seems to be a small quantitative difference, and it is a matter of semantics if it should be called ZR exciton or Hund's exciton. The authors claim the two are distinct but I could not see how they are qualitatively different.

B. The authors say that ZR scenario does not explain the way that the exciton changes with applied magnetic field quoting Ref. [18], or how it can have such a narrow line width. Does the Hund's exciton model have better explanations for these observations?

C. I agree with the authors that the F2 and F4 parameters are rather too large in Ref. [4], and ~87% of atomic values used in this work is more physical. Ref. [4] probably tried to give a good fit to RIXS and the optical data by S. Y. Kim et al. Phys. Rev. Lett. 120, 136402 (2018) at the same time. Is the authors' electronic structure also in good agreement with the transitions seen in optical data? Do the authors understand why $\Delta S=1$ transition is dipole allowed?

D. The authors attribute the feature in the high energy side to the single magnons as "double magnon". I understand this is an excitation with $\Delta S=2$, which is distinct from the usual bi-magnon. This assignment is based on the resonance behavior calculated in Fig. 4. However, the double magnon in theory resonates at a higher energy compared to the single magnon, whereas the bi-magnon resonates at a lower energy. The experimental data is close to the latter case. Although the feature around 858 eV is in better agreement with double magnon, I am not convinced that the high-energy feature should be entirely interpreted in terms of double magnons. I also suggest providing the analysis of the resonance profile in the Supplementary Information.

E. The propagation of double magnon depicted in Fig. 5 assumes that double magnon is a bound state of two single magnons, but this is not really justified in the paper. If the process in Fig. 5 was really important for the double magnon dynamics, I think there should have been some significant difference between hopping along and across the zig-zag chain, but looking at Fig. 2, there is hardly any difference.

F. In Fig. 2, the dispersions for both exciton and magnon seem almost flat except few points near $H=0$ or $K=0$. The bandwidth is much smaller than the instrumental resolution and comparable to error bars of one sigma, so I think it is difficult to claim the similarity between double magnon and exciton in terms of their dispersion. As noted above, however, I think the bigger problem is the justification of the assignment as double magnon.

Reviewer #3 (Remarks to the Author):

The manuscript presents a resonant inelastic x-ray scattering study of excitons in the van der Waals antiferromagnet NiPS₃. The authors propose that the exciton is primarily influenced by Hund's interaction and supported by cluster model calculations, challenging the previously suggested transition from a Zhang-Rice triplet to a Zhang-Rice singlet [ref. 4]. They also observe a subtle exciton dispersion and draw a parallel to double magnons. These findings enhance our understanding of excitons in NiPS₃ and align with the magnetic field-induced splitting of this excitation [ref. 18]. Nevertheless, the manuscript currently falls short of meeting the high standards of Nature Communications.

A primary concern pertains to the subtle energy dispersion of the exciton (around 15 meV), which is less than the energy resolution of 31 meV. Consequently, the conclusions rely highly on the precision of zero energy and data analysis. The authors employed a damped harmonic oscillator model to fit magnons and double-magnons and a Voigt function to fit excitons. They align the fitted magnon peak to those calculated from spin wave theory. These chosen fitting methods and model parameters introduce inherent uncertainties into the analysis. To enhance the validity of the findings, it is recommended that the authors explore alternative fitting functions for excitations to validate the energy shift. Furthermore, the manuscript could benefit from additional theoretical or computational support to substantiate the presence of dispersive excitons in a van der Waals antiferromagnet.

The authors argue that the propagation of excitons resembles double magnons, but the dispersion of double magnons appears unconvincing in Figures 2c and d. The fitted energies of double magnons exhibit significant noise, with several points randomly deviating from a flat dispersion. The explanation of the similarity between excitons and double magnons in Figure 5 is somewhat illustrative. Further clarification and evidence are required in this regard.

While the exciton remains discernible at higher temperatures, the dispersion becomes less clear. Does the exciton dispersion change with temperature?

The manuscript would benefit from the inclusion of XAS spectra. Reference 4 reports the exciton resonance at 858.1 eV, whereas the present manuscript reports an energy resonance at 853.4 eV. An elucidation for this variance is required.

Figure 1 shows an observable discrepancy between the calculated energy and experimental results, with the calculated energy consistently higher. This disparity needs to be explained.

What determines the width of the resonance behavior in Figure 4? The calculated exciton width is broader than that observed in the experiment. Furthermore, the calculation shows a difference in the satellite resonant energy between the exciton and double-bimagnon, whereas they are closer in energy in the experimental data. How to reconcile these discrepancies?

The supplementary material should include explanatory notes for supplementary figures 2 - 7 to facilitate understanding. Additionally, it is advisable to transfer some of the details of the fitting procedures in the main Methods section to the Supplementary Information to streamline the manuscript and enhance clarity.

Response to the Comments of Referee #1

Dear Reviewer #1,

Thank you for your helpful comments. We appreciate your input and are happy to hear that you consider the experimental investigations combined with theory to be convincing and suitable for publication in Nature Communications after we address your minor comments. Below we have copied your report in black and describe how we have edited the manuscript in response to your comments in blue.

1. It would be nice to add the symmetry labels for the local states to the states as one generally sees for these local atomic multiplets.

Thank you for this suggestion. We have added the symmetry labels in Fig. 1c for these states. This was done by computing the numerical overlap between our derived eigenvectors with the group-theory results in *Multiplets of Transition-Metal Ions in Crystals* by Tanabe, Sugano, & Kamimura. We enlisted an additional person in the collaboration to do this, Mikołaj Walicki, and he has been added to the author list.

2. The picture of Fig 1.h shows the local “wave-function” or hole charge density. The colour indicates the spin of the state. It is not clear how this picture is made. At the same time I do not understand the change in spin between the Ni site and the O site. If these are the results of a Ni-Ligand cluster calculation I would expect the spin to be all parallel. (for the ground-state).

Thank you for helping us clarify this point. All these values indeed come from the Ni-Ligand cluster calculation. The size of the Ni 3d and S 3p orbitals is proportional to their hole occupation, which was obtained by computing the expectation value of the hole operator for each state separately. Regarding spin, we computed and the expectation value of the spin operator along the z-axis $\langle \hat{S}_z \rangle$, again separately for the Ni 3d and S 3p orbitals. Both Ni and S sites have negative $\langle \hat{S}_z \rangle$, so the spins are parallel in this sense, but the Ni states host a larger magnitude of spin.

To improve the clarity of the manuscript, we have added the following sentence in the caption of Fig. 1:

“The size of each orbital (3d for the central Ni site and 3p for the six neighboring S sites) is proportional to its hole occupation. The color represents the expectation value of the spin operator along the z axis $\langle \hat{S}_z \rangle$, again calculated separately for the Ni and S states.”

Response to the Comments of Referee #2

Dear Reviewer #2,

Thank you for your careful review of our manuscript. Below we have copied your report in black and describe how we have edited the manuscript in response to your comments in blue. In cases where your points contain several different questions/comments we use **bold** subheadings.

1. Since the charge transfer energy is positive, it is clear that the ground state has a larger weight for d^8 relative to d^9L^{-1} , in both this work and Ref. [4]. The difference between these two works in terms of the hole occupation on the Ni Site is 1.26 versus 1.08. To me, this seems to be a small quantitative difference, and it is a matter of semantics if it should be called ZR exciton or Hund's exciton. The authors claim the two are distinct but I could not see how they are qualitatively different.

Thank you for raising this important question, which gives us an important opportunity to further clarify the meaning and importance of our work. The Zhang-Rice character that was previously claimed to describe the exciton comes from the eponymous paper by Zhang and Rice, Physical Review B 37, 3759–3761 (1988) [1] and is not just a difference in which atoms hosts the holes, but a specific form for the wavefunctions involved in the transition. In our work, we find that the Zhang-Rice wavefunctions are *not* the leading component of the ground state wavefunction, [see Eq. (6) of SI]. Correctly describing the wavefunctions has major implications for the interactions that stabilize the exciton and future efforts to manipulate the exciton.

We have made small changes to the introduction and discussion to draw more attention to this fact, stating

“This terminology derives from studies of cuprate superconductors, and refers to a specific form of hybridized wavefunctions that have one hole on the transition metal (Ni) site and one hole on the ligand (S) site and describes the “Zhang-Rice exciton” as a transition from a high-spin triplet to a low-spin singlet [1].”

and in the Discussion Section

“As we discuss later, the distinction between these models is crucial as it corresponds to a different majority component of the wavefunction, different interactions playing the leading role in the exciton energy, and the possibility of realizing a model with physically reasonable parameters. These issues will be central to efforts to manipulate the exciton energy and cross section.”

We have added further discussion points as follows:

“It also implies that future efforts to realize similar symmetry excitons with different energies should target means to modify the on-site Ni Hund's exchange coupling and

not the Ni-S exchange processes that would be the leading contribution to the energy of a Zhang-Rice exciton.”

2. The authors say that ZR scenario does not explain the way that the exciton changes with applied magnetic field quoting Ref. [18], or how it can have such a narrow line width. Does the Hund’s exciton model have better explanations for these observations?

A Hund’s exciton picture, as we advocate here, is essentially “an internal excitation within the $d-d$ states of Ni^{2+} ions” as speculated in Ref. 18. Therefore, it is compatible with a magnetic field dependence, and would address the assertion made previously in Ref. 18. The Hund’s exciton picture is also more naturally compatible with the observed narrow line width (i.e., long lifetime), because this is in some sense the simplest way to realize an on-site triplet-singlet excitation on Ni $3d$ orbitals. There are relatively few excitations in NiPS_3 and the exciton is energetically well separated from other transitions, such that it interacts exclusively by magnons and not other types of excitation. The wavefunction for the Hund’s exciton is also very localized on Ni and mostly due to an on-site Hund’s spin flip, which provides a natural possible explanation for why it does not interact strongly with phonons and decay (see the discussion below). This draws parallels with prior suggestions that the NiPS_3 exciton comes from extrinsic atomic defects [2] in the sense that the Hund’s exciton (while being intrinsic) predominantly exists on a single Ni atom. We have expanded one of the discussion points to address this.

“Consequently, NiPS_3 has relatively few excitations compared to other materials and the exciton is energetically well separated from other transitions, such that it interacts exclusively by magnons and not other types of excitations. The exciton character is also dominated by processes that rearrange spins on the Ni site, rather than moving charge between more extended states, which would tend to reduce the coupling of the exciton to phonons. These factors may contribute to the long lifetime and narrow linewidth of the exciton.”

3. I agree with the authors that the F2 and F4 parameters are rather too large in Ref. [4], and 87% of atomic values used in this work is more physical. Ref. [4] probably tried to give a good fit to RIXS and the optical data by S. Y. Kim et al. Phys. Rev. Lett. 120, 136402 (2018) at the same time. Is the authors’ electronic structure also in good agreement with the transitions seen in optical data? Do the authors understand why $\Delta S = 1$ transition is dipole allowed?

We thank the reviewer for these constructive questions.

Comparison with optical results. Our results are consistent with the optical data. In the optical conductivity spectra reported in the work by S. Y. Kim et al. [3], there are absorption peaks at 1.1, 1.7, 2.2, 3.5, and 4.6 eV, which exactly match the excitations in both our experimental and calculated RIXS spectra (cluster model or the equivalent AIM) as shown in Supplementary Fig. 2. On the contrary, the calculated spectra based on the model in Ref. [4] in the paper (Entry [4] in the References section of this document) cannot reproduce the energies of the last three excitations well (see Supplementary Fig. 2e). In fact, the model proposed in Ref. [3] was based on XAS data, which contains less information than the RIXS spectra and cannot determine all the model

parameters unambiguously. In Ref. [4], the model was claimed to be solely obtained from fitting their RIXS spectra without considering the previous optical data. In contrast to the original model, the updated model in Ref. [4] actually gives a positive charge transfer energy instead of the initially claimed negative value in Ref. [3]. Our model more faithfully captures the detailed splitting of the two dd -excitations at 1.0 and 1.1 eV in our high-resolution RIXS spectra, is more consistent with the previous optical data, and has more physically reasonable values for the Coulomb interaction parameters. We have added the following comment on this in the Supplementary Note 1F when we compare different models:

“Moreover, our model is compatible with the previous optical work, which reported absorption peaks at 1.1, 1.7, 2.2, 3.5, and 4.6 eV [3]. On the contrary, the calculated spectra using the model in Ref. A [4] cannot reproduce the energies of the last three excitations well.”

$\Delta S = 1$ transition. We believe that the presence of a nominally dipole-forbidden process in the optical spectra, is closely related to the same exciton-spin interactions that we study here. We have added further discussion of this to the manuscript:

“The coupling between the exciton and magnetism might be relevant to why the 1.47 eV exciton feature is visible in optics experiment. Since the exciton involves transitions between d -orbitals, it is expected to be nominally optically forbidden in a centrosymmetric crystal due to dipole selection rules. However, these rules can be lifted by perturbations that break the Ni-site symmetry, which includes exciton-spin or exciton-lattice interactions. The fact that there is a strong change in the optical cross section through T_N [3], whereas the RIXS is only modestly broadened, also supports the interpretation that exciton-spin interactions play a key role in the optical cross section.”

4. The authors attribute the feature in the high energy side to the single magnons as “double magnon”. I understand this is an excitation with $\Delta S = 2$, which is distinct from the usual bi-magnon. This assignment is based on the resonance behavior calculated in Fig. 4. However, the double magnon in theory resonates at a higher energy compared to the single magnon, whereas the bi-magnon resonates at a lower energy. The experimental data is close to the latter case. Although the feature around 858 eV is in better agreement with double magnon, I am not convinced that the high-energy feature should be entirely interpreted in terms of double magnons. I also suggest providing the analysis of the resonance profile in the Supplementary Information.

We appreciate your input on this point. We agree that the main resonance around 853 eV shows overlap between the double-magnon and bi-magnon processes, separated by only one energy point in a way that is very sensitive to the details of the data, the fitting, and the model. However, if the full experimental spectrum including the resonances around 853 eV and 857.7 eV, is considered, we can see that the excitation in question is present at both resonances 853 eV and 857.7 eV. This property is compatible with a double-magnon and incompatible with a bi-magnon. We use this property as the basis for the assignment of the leading character of the excitation. We did not intend to give the impression that the bi-magnon intensity must be precisely zero. Our assignment

is also supported by RIXS studies of NiO where the double-magnon was also found to be much more intense than the bi-magnon [5, 6, 7, 8]. We have improved the relevant section of the manuscript:

“The main resonance around 853 eV shows partial overlap between the double-magnon and bi-magnon resonance, so it does not clearly distinguish between them. However, the presence of the excitation at the satellite resonance at 857.7 eV is only compatible with a $\Delta m_S = 2$ double-magnon process. We therefore suggest that the dominant character of the excitation around 80 meV is that of double-magnons substantiating our spectral assignment, although we cannot exclude a small sub-leading contribution. Our assignment is also supported by RIXS studies of NiO where the double-magnon was also found to be much more intense than the bi-magnon [5, 6, 7, 8].”

We have also added an expanded section for resonance behavior analysis in Supplementary Note 3F and Supplementary Figs. 11–13.

5. The propagation of double magnon depicted in Fig. 5 assumes that double magnon is a bound state of two single magnons, but this is not really justified in the paper. If the process in Fig. 5 was really important for the double magnon dynamics, I think there should have been some significant difference between hopping along and across the zig-zag chain, but looking at Fig. 2, there is hardly any difference.

We appreciate the reviewer for raising this question.

Anisotropy in the double-magnon dispersion. There are two effects that lead to the similarity between the two in-plane directions. The first is that the leading J_3 exchange process connects only antiferromagnetically aligned Ni spins, and the second is that the sample is structurally and magnetically twinned. We have added the following sentences elaborating these points:

In the Supplementary Information:

“We note that the leading J_3 exchange process connects only antiferromagnetically Ni spins, which would suppress any difference between propagation along different directions in the lattice. This is in addition to the fact that NiPS₃ is structurally and magnetically twinned, meaning that the ferro- and antiferro-magnetic directions in the lattice are not empirically distinguishable (see the methods section).”

In the main text:

“The apparent similarity in the measured dispersion between the two distinct directions (i.e., antiferromagnetic across the zig-zag chain and ferromagnetic along the chain) can be ascribed to the domain averaging effect due to the presence of structural twinning in the measured sample.”

Bound-state assumption for the double-magnon. It is correct to say that we do not have explicit proof that the double-magnon excitation is a bound state. This was stated in the

supplementary information of the manuscript. Due to your helpful comments, we have extended the paragraph in question with some sentences about studying monodomains of NiPS₃.

“These estimations for the exchange process have implicitly assumed that the double-magnon is a bound state, which is formally justified only in the limit of large Ising anisotropy. For NiPS₃, the double-magnon probably has high decay rates into two nearest-neighbor single-magnon states. Fortunately, the latter should remain bound (due to attractive interactions between magnons on nearest neighbor sites), so our analysis should remain a good order-of-magnitude estimate. . . . Although technically very challenging, the development of ultrahigh energy resolution RIXS under strain could be implemented to study specific magnetic monodomains to more directly test whether the double-magnon is or is not a bound state. ”

Pairs of magnon excitations will generally experience effective attractive interactions due to the fact that magnons break fewer magnetic bonds when they are in close proximity.

6. In Fig. 2, the dispersions for both exciton and magnon seem almost flat except few points near $H=0$ or $K=0$. The bandwidth is much smaller than the instrumental resolution and comparable to error bars of one sigma, so I think it is difficult to claim the similarity between double magnon and exciton in terms of their dispersion. As noted above, however, I think the bigger problem is the justification of the assignment as double magnon.

We would agree that the effects are subtle, but we are confident of the dispersions we extracted for both the exciton and double-magnon. First of all, the error bars after taking into account all possible sources are still smaller than the bandwidth as shown in Fig. 2. Secondly, the dispersions matches the periodicity of the Brillouin zone in both H and K directions, i.e., consistently lower in energy near Brillouin zone center and maximized near Brillouin zone boundaries. Next, we did careful cross-checks. For the exciton, the Brillouin zone center energy is consistent with the values from previous optical measurement; the fitted magnon and elastic peak intensities match the expectation; and the null hypothesis test of a flat exciton dispersion leads to unphysical results. Similarly for the double-magnon, the null hypothesis test also gives bad fits; and using a different functional form in fitting the magnon and double-magnon peaks result in quantitatively similar dispersions. In the last, the fit to the dispersions based on an empirical tight-binding model shows that the third nearest neighbor interaction is dominant, consistent with our expectation that exciton hopping and spin exchange processes are similar. Here, the null-hypothesis test for the double-magnon dispersion (Supplementary Note 3C and Supplementary Fig. 7), the Voigt fits to the magnon and double-magnon peaks (Supplementary Note 3D and Supplementary Fig. 8), and the tight-binding model fit (Supplementary Note 5 and Supplementary Fig. 14) are newly added in Supplementary Information.

References

- [1] Zhang, F. C. & Rice, T. M. Effective Hamiltonian for the superconducting Cu oxides. *Physical Review B* **37**, 3759–3761 (1988).

- [2] Kim, D. S. *et al.* Anisotropic excitons reveal local spin chain directions in a van der Waals antiferromagnet. *Advanced Materials* **35**, 2206585 (2023).
- [3] Kim, S. Y. *et al.* Charge-Spin Correlation in van der Waals Antiferromagnet NiPS₃. *Physical Review Letters* **120** (2018).
- [4] Kang, S. *et al.* Coherent many-body exciton in van der Waals antiferromagnet NiPS₃. *Nature* **583**, 785–789 (2020).
- [5] de Groot, F. M. F., Kuiper, P. & Sawatzky, G. A. Local spin-flip spectral distribution obtained by resonant x-ray Raman scattering. *Phys. Rev. B* **57**, 14584–14587 (1998).
- [6] Ghiringhelli, G. *et al.* Observation of two nondispersive magnetic excitations in NiO by Resonant Inelastic Soft-X-Ray Scattering. *Phys. Rev. Lett.* **102**, 027401 (2009).
- [7] Betto, D. *et al.* Three-dimensional dispersion of spin waves measured in NiO by resonant inelastic x-ray scattering. *Physical Review B* **96**, 020409 (2017).
- [8] Nag, A. *et al.* Many-body physics of single and double spin-flip excitations in NiO. *Physical Review Letters* **124**, 067202 (2020).

Response to the Comments of Referee #3

Dear Reviewer #3,

Thank you for your helpful review. We agree that our manuscript enhances our understanding of these highly topical NiPS₃ excitons and have incorporated your suggestions to make several major improvements to the text. We believe that these changes significantly improved the quality of the manuscript at a level suitable for Nature Communications. Below we have copied your report in black and describe how we have edited the manuscript in response to your comments in blue. In cases where your points contain several different questions/comments we use **bold** subheadings.

1. A primary concern pertains to the subtle energy dispersion of the exciton (around 15 meV), which is less than the energy resolution of 31 meV. Consequently, the conclusions rely highly on the precision of zero energy and data analysis. The authors employed a damped harmonic oscillator model to fit magnons and double-magnons and a Voigt function to fit excitons. They align the fitted magnon peak to those calculated from spin wave theory. These chosen fitting methods and model parameters introduce inherent uncertainties into the analysis. To enhance the validity of the findings, it is recommended that the authors explore alternative fitting functions for excitations to validate the energy shift. Furthermore, the manuscript could benefit from additional theoretical or computational support to substantiate the presence of dispersive excitons in a van der Waals antiferromagnet.

We thank the reviewer for these constructive comments.

Fitting method. As long as we have good quality of data, we should be able to easily determine a peak position with a precision well below the instrument resolution. Indeed, for the exciton peak, the bare error bars from our fits are only around 0.5–1 meV. The double-magnon peak is broader and closer to the strong magnon peak, so the bare error bars are slightly larger, around 3 meV, but still well below the bandwidth. The main contribution, as pointed by this reviewer, is from the determination of the zero energy. As explained in the manuscript and supplementary materials, we estimated this uncertainty and included it in the error bars. We further cross-verified this procedure by comparing to both inelastic neutron scattering and optics data.

Following your helpful suggestion, we have explored an alternative fitting function (i.e., Voigt function instead of the damped harmonic oscillator model) for the magnon and double-magnon peaks. The new result is similar to the previous fits, validating the robustness of the exciton and double-magnon dispersions we observed. This has been added as a new section (Supplementary Note 3D and Supplementary Fig. 8) in Supplementary Information.

Theoretical support of dispersive excitons. A full ab initio theory for dispersive Hund's excitons, of the type one might ideally hope for, is well beyond what is currently feasible. Indeed, the current manuscript contains a comment that state-of-the-art Bethe-Salpeter calculations fail to account for the exciton dispersion, very likely due to the correlated nature of the Hund's exciton [1].

In response to your question, we developed a tight-binding model to describe the exciton dispersion. This is widely used to interpret exciton dispersions in molecular solids, despite being a

simple effective-parameter-based approach [2]. The results indicate that the third nearest neighbor process is the dominant interaction in the exciton motion. This is consistent with the fact that the third nearest neighbor spin exchange is the dominant term in the spin Hamiltonian of NiPS₃ and our expectation that exciton hopping and spin exchange processes are similar in this material. This has been added as a new section (Supplementary Note 5 and Supplementary Figure 14) in Supplementary Information with the following sentence in the main text:

“A simple empirical tight-binding model fit to the exciton dispersion (see Supplementary Note 5) reveals that the third nearest neighbor interaction is the leading term in determining the exciton dispersion, consistent with the third nearest neighbor spin exchange being the dominant term in the spin Hamiltonian [3, 4]. ”

2. The authors argue that the propagation of excitons resembles double magnons, but the dispersion of double magnons appears unconvincing in Figures 2c and d. The fitted energies of double magnons exhibit significant noise, with several points randomly deviating from a flat dispersion. The explanation of the similarity between excitons and double magnons in Figure 5 is somewhat illustrative. Further clarification and evidence are required in this regard.

We thank the reviewer for these constructive comments.

Double-magnon dispersion. We agree that the dispersion effects are subtle. However, the bandwidth of the dispersion is still larger than the error bars. The deviation from a flat dispersion happens not “randomly”, but mainly near the Brillouin zone center, similar to the exciton dispersion. To further corroborate this dispersion, we have added a new analysis assuming the null hypothesis of a flat dispersion for the double-magnons, which has been added into Supplementary Note 3C and Supplementary Figure 7. This null hypothesis can be safely excluded by examining the fitted lines, which clearly deviate from the best fits and cannot describe the experimental data well.

Figure 5. Figure 5 is an illustration, but it is one that depicts the leading processes in the perturbation theory describing the exciton and double-magnon motion. As such, it represents the theoretical expectations of how these quasiparticles are expected to move. The complete theory has been detailed in Supplementary Note 3 (now Supplementary Note 4). We have updated the caption of Fig. 5 to make the relevance of the figure more immediately evident, such that the caption now reads:

“Illustration of exciton and double-magnon propagation based on perturbation theory.”

3. While the exciton remains discernible at higher temperatures, the dispersion becomes less clear. Does the exciton dispersion change with temperature?

We thank the reviewer for this insightful question. As we mentioned in the Method Section (which now has been moved to Supplementary Information, see point 7 below), due to the increased error bars for the energy zero determination and the broadening of the exciton peak, the error bars for the fitted exciton peak positions are much larger at 190 K. This increased uncertainty prevents us from drawing any decisive conclusions of the exciton dispersion at this temperature.

In response, we have added the following sentence in the main text:

“Above T_N the exciton remains visible, but it becomes weaker and more diffuse compared to the data at 40 K (see Fig. 3a and the linecuts in Supplementary Fig. 10). Consequently, no dispersion is detectable.”

4. The manuscript would benefit from the inclusion of XAS spectra. Reference 4 reports the exciton resonance at 858.1 eV, whereas the present manuscript reports an energy resonance at 853.4 eV. An elucidation for this variance is required.

We thank the reviewer for the suggestion and have included an XAS spectra as Fig. 1 in Supplementary Information. As for the shift of the resonance energy, this is related to the absolute energy calibration of the x-ray beamline. We believe our calibration is more accurate. The main L_3 -edge resonant energy obtained from our measurements is 852.8 eV, which is within the common range of the Ni L_3 -edge energy (around 852–854 eV) from XAS measurements on other Ni containing compounds [5, 6, 7]. With that being said, what matters here is the relative energy instead of the absolute value. Therefore, if we compare the relative position of the exciton resonance peak, our observation is consistent with the result reported in Reference 4.

To improve the clarity of the manuscript, we have added the following sentence in Methods Section:

“The main resonance energy (around 853 eV) is common for Ni-containing compounds [5, 6, 7] but different from the previous report [8]. This difference comes from the absolute energy calibration of the beamline, but does not affect the RIXS measurements and interpretations, which depend only on the relative changes. ”

5. Figure 1 shows an observable discrepancy between the calculated energy and experimental results, with the calculated energy consistently higher. This disparity needs to be explained.

We thank the reviewer for pointing this out. There was an error in what was plotted due to the added Zeeman term (see Magnetic cross-section calculations in Methods Section) on top of the refined model. This term was included to simulate the magnetically ordered phase in which the ground state triplet splits. However, we realized that this added spin exchange should be applied only to the ground state instead of all the excitations. We have turned off this Zeeman term for the excited states in the updated Fig. 1, and now the calculated energies match the experimental results well as expected.

6. What determines the width of the resonance behavior in Figure 4? The calculated exciton width is broader than that observed in the experiment. Furthermore, the calculation shows a difference in the satellite resonant energy between the exciton and double-bimagnon, whereas they are closer in energy in the experimental data. How to reconcile these discrepancies?

Peak width. The width of a resonance peak reflects the lifetime of the XAS final state, which is primarily determined by the properties of the Ni $2p_{3/2}$ core hole, but also weakly dependent

on the precise valence configuration. In our model calculations, we used a single inverse core-hole lifetime for all the valence configurations and determined its value ($\Gamma_c = 0.6$ eV half-width at half-maximum) by fitting to the observed width of the main resonance peak in the XAS spectra, which was set mainly based on the width of the strongest excitations around 1.1 eV and 1.7 eV. This is standard practice for the vast majority of RIXS calculations, mainly because the core hole lifetime offers very few insights into the low-energy physics of quantum materials, but also because it is difficult to calculate. Other excitations, including the Hund's exciton, could have shorter or longer core-hole lifetime compared to this median value, which is why these small differences are visible. It would be feasible to develop a more complex model, where different width parameters are used for each excitation. However, we do not consider this helpful since it would not bring new insights to the problem and it would be contrary to standard practice in analyzing this type of experiment.

Positions for the satellite peaks. We do observe subtle differences between the experimental and calculated peak positions of the satellite peaks. The satellite peaks appear naturally in our cluster model (or the equivalent AIM) after we fit the main resonance features in the RIXS spectra. The small differences between theory and experiment could be due to the limited number of free parameters in the model. This is evidenced even in the first RIXS report of the double-magnon excitations in the prototypical material NiO, where the calculated satellite peak position is also slightly deviated from the experimental results (Fig. 3 in Ref. [9]). However, we want to emphasize that the current simple model can already generate satisfactory results in terms of simulating the existence of the satellite peaks. The calculated resonance energy is also reasonably close to the experimental value.

7. The supplementary material should include explanatory notes for supplementary figures 2 - 7 to facilitate understanding. Additionally, it is advisable to transfer some of the details of the fitting procedures in the main Methods section to the Supplementary Information to streamline the manuscript and enhance clarity.

We thank the reviewer for this suggestion. We have added a new section named "Supplementary Note 3. Detailed fitting procedures for the RIXS spectra" in Supplementary Information, and moved most of the details of the fitting procedures in Methods Section to there. This new section also contains explanatory notes for Supplementary Figs. 2–7 in the previous version.

References

- [1] Lane, C. & Zhu, J.-X. An ab initio study of electron-hole pairs in a correlated van der waals antiferromagnet: NiPS₃ (2022). arXiv:2209.13051.
- [2] Cudazzo, P., Sottile, F., Rubio, A. & Gatti, M. Exciton dispersion in molecular solids. *Journal of Physics: Condensed Matter* **27**, 113204 (2015).
- [3] Scheie, A. *et al.* Spin wave hamiltonian and anomalous scattering in NiPS₃. *Phys. Rev. B* **108**, 104402 (2023).
- [4] Wildes, A. R. *et al.* Magnetic dynamics of NiPS₃. *Physical Review B* **106**, 174422 (2022).

- [5] Nakamura, T., Oike, R., Ling, Y., Tamenori, Y. & Amezawa, K. The determining factor for interstitial oxygen formation in Ruddlesden–Popper type La_2NiO_4 -based oxides. *Phys. Chem. Chem. Phys.* **18**, 1564–1569 (2016).
- [6] Hepting, M., Dean, M. P. M. & Lee, W.-S. Soft X-Ray Spectroscopy of Low-Valence Nickelates. *Frontiers in Physics* **9** (2021).
- [7] Wang, H. *et al.* Nickel L-Edge Soft X-ray Spectroscopy of Nickel-Iron Hydrogenases and Model Compounds – Evidence for High-Spin Nickel(II) in the Active Enzyme. *Journal of the American Chemical Society* **122**, 10544–10552 (2000).
- [8] Kang, S. *et al.* Coherent many-body exciton in van der Waals antiferromagnet NiPS_3 . *Nature* **583**, 785–789 (2020).
- [9] Nag, A. *et al.* Many-body physics of single and double spin-flip excitations in NiO . *Physical Review Letters* **124**, 067202 (2020).

REVIEWERS' COMMENTS

Reviewer #1 (Remarks to the Author):

The authors have answered all my questions and in my opinion the manuscript can be published in its current form.

Reviewer #2 (Remarks to the Author):

In my previous report, my main concern was that there is only a small quantitative difference between the "correct" wavefunction claimed in this study and the one that was proposed in Ref. [4]. I understand the point that the ground state has a dominant d8 character and ZR state is not the leading component. But this was already clear from the fact that the charge transfer energy is positive. If the authors' point is that the exciton should be called a "Hund's exciton", I have no problem with that. But the problem is that just giving it a different name does not bring any new physics. After all, the symmetry of the wavefunctions are the same, and the difference is just that this study finds 1.26 hole on the Nickel site as compared to the 1.06 in the previous study. I do not see what qualitative difference it brings, or at least the authors have failed to show that it does.

Looking at the RIXS dispersion, whether or not the dispersion is corrected extracted by an accurate fitting is a secondary question, what is more problematic is that there is no experimental evidence whatsoever that the exciton dispersion is renormalized by background AF spins.

So all in all, there is really no fundamentally new stuff that is offered in this paper. Therefore, I am not able to recommend publication of this paper in Nature communications.

Reviewer #3 (Remarks to the Author):

The authors have diligently addressed and incorporated all of my comments, significantly improving the manuscript's quality and clarity. I am pleased to recommend its publication in Nature Communications.

Response to the Comments of Referees

Below we have copied the report in black and provide responses in blue.

Response to the Comments of Reviewer #1

Dear Reviewer #1,

Thank you for your helpful input, we are very happy to hear that we have answered all of your questions and that the manuscript can be published in its current form.

Response to the Comments of Reviewer #2

Dear Reviewer #2,

Thank you for your efforts to strengthen the manuscript. Below we provide some responses to your input. Unfortunately, we felt that some of the comments included incorrect and unexplained assertions, which made it difficult to respond to these in the spirit of a constructive dialog.

Regarding Hund’s vs. Zhang-Rice exciton character: I understand the point that the ground state has a dominant d8 character and ZR state is not the leading component. But this was already clear from the fact that the charge transfer energy is positive. If the authors’ point is that the exciton should be called a “Hund’s exciton”, I have no problem with that. But the problem is that just giving it a different name does not bring any new physics. After all, the symmetry of the wavefunctions are the same, and the difference is just that this study finds 1.26 hole on the Nickel site as compared to the 1.06 in the previous study. I do not see what qualitative difference it brings, or at least the authors have failed to show that it does.

We agree that it is possible to see problems with the stated conclusions in Ref. [4] through an expert analysis of some of the contradictions within the article. As well as the problem with the charge-transfer energy Δ , the Hund’s interactions used are unphysically large. We fully accept that this is clear to you. Inspecting the literature, however, it is obvious that this point is not clear to the scientific community.

Ref. [4] asserts their conclusions of a Zhang-Rice exciton very clearly saying “we determine the origin of the coherent excitonic excitation to be a transition from a Zhang–Rice triplet to a Zhang–Rice singlet”. This is in the abstract of a highly influential recent Nature paper from 2020 with 184 citations as of today. Many of the citations directly parrot the Zhang–Rice character of the NiPS₃ exciton. Additional papers have appeared on other materials misleadingly claiming that they have Zhang-Rice excitons too, e.g. work on NiI₂ [1]. So calling it a “Hund’s exciton” is not simply giving it a “different” name but a correct name that properly describes the true physical process, which will be beneficial to the whole community.

We agree that Zhang-Rice and Hund’s excitons share the same symmetry, but we strongly disagree that this implies that our conclusion of a Hund’s exciton is uninteresting. Many highly

interesting situations in physics involve states that share the same symmetry. Mott versus charge-transfer insulator would be a pertinent example. The manuscript also contains extensive discussion of why this distinction is important. For example, the majority component of the exciton needs to be understood to know how to manipulate the exciton energy (which is key to potential applications). These efforts should target means to modify the on-site Ni Hund's exchange coupling and not the Ni-S exchange processes that would be the leading contribution to the energy of a Zhang-Rice exciton.

The spatial extent of the Hund's exciton is also very localized on Ni and mostly due to an on-site Hund's spin flip. This factor is important because it provides a natural possible explanation for why it does not interact strongly with phonons and decay. This draws parallels with prior suggestions that the NiPS₃ exciton comes from extrinsic atomic defects [2] in the sense that the Hund's exciton (while being intrinsic) predominantly exists on a single Ni atom.

The form of the wavefunction is also central to the superexchange processes that control how the exciton disperses.

Looking at the RIXS dispersion, whether or not the dispersion is corrected extracted by an accurate fitting is a secondary question, what is more problematic is that there is no experimental evidence whatsoever that the exciton dispersion is renormalized by background AF spins.

We respectfully but emphatically disagree with the assertion that there is no experimental evidence whatsoever that the exciton dispersion is renormalized by background AF spins. In our manuscript, we show that the exciton dispersion mirrors the double-magnon dispersion. This is exactly the direct experimental evidence that the exciton dispersion is renormalized by background AF spins that the referee is asserting is not in the paper. One can look at other papers such as Ref. [3] to see other times that the similarity of an exciton dispersion and spin-dispersion were compared to infer spin-interactions with an exciton. For this comment to be scientifically useful, the report would need to include a suggestion for a different type of evidence for a magnetically doped exciton. We, personally, are not aware of a more appropriate experimental approach than the one we used.

Response to the Comments of Reviewer #3

Dear Reviewer #3,

Thank you for your constructive reviewing, we are delighted to hear that we have answered all of your questions and that the manuscript can be published in its current form.

References

- [1] Son, S. *et al.* Multiferroic-enabled magnetic-excitons in 2D quantum-entangled van der Waals antiferromagnet NiI₂. *Advanced Materials* **34**, 2109144 (2022).
- [2] Kim, D. S. *et al.* Anisotropic excitons reveal local spin chain directions in a van der Waals antiferromagnet. *Advanced Materials* **35**, 2206585 (2023).

- [3] Kim, J. *et al.* Excitonic quasiparticles in a spin-orbit Mott insulator. *Nature Communications* **5** (2014).